# Anticancer properties of peptides and protein hydrolysates derived from Asian water monitor (*Varanus salvator*) serum

Jitkamol Thanasak[1]*, Sittiruk Roytrakul[2], Rudee Surarit[3,4], Waraphan Toniti[5], Wanna Sirimanapong[1], Janthima Jaresitthikunchai[2], Narumon Phaonakrop[2], Siriwan Thaisakun[2], Sawanya Charoenlappanit[2], Surasak Jittakhot[5]

**1** Department of Clinical Sciences and Public Health, Faculty of Veterinary Science, Mahidol University, Nakhon Pathom, Thailand, **2** Functional Proteomics Technology Laboratory, National Center for Genetic Engineering and Biotechnology (BIOTEC), National Science and Technology Development Agency, Pathum Thani, Thailand, **3** Department of Oral Biology, Faculty of Dentistry, Mahidol University, Bangkok, Thailand, **4** Faculty of Dentistry, Siam University, Bangkok, Thailand, **5** Department of Preclinic and Applied Animal Science, Faculty of Veterinary Science, Mahidol University, Nakhon Pathom, Thailand

* jitkamol.tha@mahidol.ac.th

## Abstract

This study investigated the anticancer efficacy of <3 kDa fractions derived from native peptides and protein hydrolysate of *Varanus saltator* serum. The inhibitory effects of these fractions were evaluated against a panel of cancer cell lines (A375, CaCO2, CAL27, NCI-H460, HeLa, HCT8, HT29, HepG2, KATO III, MCF-7, MDA-MB-231, Raw264.7, SKOV-3, SW620, T47D, and U937) and normal cell lines (HaCaT, MRC5, and Vero). Native peptides demonstrated higher anticancer activity compared to protein hydrolysates, inhibiting 16 cell lines and exhibiting high efficacy (≥70% inhibition) against CaCO2, CAL27, HaCaT, HT29, HepG2, MCF-7, MRC5, and U937. These native peptides were further fractionated by stepwise reverse-phase column chromatography. The hydrophilic (C18 unbound) peptide fraction exhibited greater anticancer activity than the hydrophobic (C18 bound) fraction. In addition, by LC-MS analysis, the peptide sequences were screening *in silico*. The predictions showed that 159 of the 432 Varanus peptides had the potential to be anticancer peptides (ACPs), of which the top twenty had a probability of more than 75%. The anticancer mechanism of peptides may be explained by the mechanism of cell entry or action. Further peptide synthesis and modification should be the next step to enhance the anticancer efficacy of these peptides with less toxicity to Vero cells. This finding sets the way for the development of new anticancer drugs originating from *Varanus salvator* serum peptides.

## Introduction

Cancer remains a leading global health concern, with both incidence and mortality rates continuing to increase. In 2022, there were an estimated 20 million new cancer cases and 9.7 million cancer-related deaths worldwide. This reflects a significant increase from previous years, highlighting the growing burden of cancer globally [1]. Lung cancer is the most commonly

**Data availability statement:** All relevant data are within the manuscript and its Supporting Information files.

**Funding:** The research project was supported by Mahidol University (Award number: NDFR 45/2565 Recipient: Jitkamol Thanasak). The funders had no role in study design, data collection and analysis, decision to publish, or preparation of the manuscript.

**Competing interests:** The authors have declared that no competing interests exist.

diagnosed cancer and the leading cause of cancer death overall and in men worldwide, with almost 2.5 million cases and 1.8 million deaths. In females, breast cancer is the most commonly diagnosed cancer and the leading cause of cancer death, followed by lung, colorectal, and cervical cancers [2].

Conventional cancer treatments, such as surgery, chemotherapy, and radiation therapy, are often effective but come with significant limitations. Chemotherapy and radiation often cause side effect such as damage to healthy cells, metastatic, recurrence, drug toxin, organ damage etc. So, the studies on antimicrobial peptides and anticancer peptides have become popular in recent years. Anticancer peptides are thought to be an alternative treatment that are likely to be more efficient than chemical agents and cause fewer side effects. Peptides are a new group of biomolecules that are effective at inhibiting the growth of resistant and nonresistant pathogenic bacteria, including various types of cancer cells [3–12]. Previous reports indicate that antimicrobial peptides can be useful for anticancer therapeutic purposes. For example, an antimicrobial peptide synthesized from CAMEL (KWKLFK KIGAVLKVL-NH$_2$) has anticancer potential for B16-F10 murine melanoma tumors [13]. Warnericin RK (WRK) and its derivatives (WarnG20D and WarnF14V) were found to be cytotoxic to leukemic cells (Jurkat, KG1, KS62), prostate and glial cancer cells, prostate healthy cells and astrocytes [14].

Natural sources of bioactive peptides with anticancer effects have been reviewed in plants and animals. It has been reported that the anticancer potential of natural compounds such as coumarin may modulate various signaling pathways, including those involved in cell cycle regulation, apoptosis, and angiogenesis. In addition, some studies highlight the synergistic effects of combining coumarins with anticancer peptides or other therapeutic agents to enhance cancer treatment efficacy [15–20]. Animal sources of bioactive peptides with anticancer effects including terrestrial mammals and byproducts, milk and dairy products, marine animals, amphibians, animal venoms and reptiles have been reported [21,22]. The anticancer potential of reptile-derived components has been reviewed for several species, such as geckos, turtles, snakes, scorpions, crocodiles and Asian water monitors [21]. For example, two HPLC-separated mucous fractions (F2 and F5) derived from giant African snails (*Achatina fulica*) showed *in vitro* cytotoxicity against a breast cancer cell line (MCF-7) [23]. A peptide extracted from scorpion venom, BmKn2, has been shown to inhibit bacterial and cancer cell growth [10,24]. The anticancer properties of Siamese crocodile (*Crocodylus siamensis*) blood have been investigated both *in vitro* and *in vivo*. The modified cationic antimicrobial peptides KT2 and RT2 derived from *Crocodylus siamensis* Leucrocin I exhibited anticancer activity against human colon cancer HCT-116 cells, in which RT2 peptide upregulated proteins, including CFTR, Wnt7a, TIA1, PADI2, NRBP2, GADL1, LZIC, TLR6, and GPR37, within the tumor, resulting in suppressed tumor growth in mice [25–27].

In addition, investigations of reptilian serum samples indicated that the presence of Asian water monitor (*Varanus salvator*), python (*Malayopython reticulatus*) and tortoise (*Cuora kamaroma amboinensis*) inhibited the viability of Henrietta Lacks cervical adenocarcinoma cells (HeLa), prostate cancer cells (PC3) and human breast adenocarcinoma cells (MCF7 cells). The potential anticancer peptides (ACPs) were 123/883 peptides from *V. salvator*, 306/1074 peptides from *Malayopython reticulatus* and 235/885 peptides from *C. kamaroma amboinensis* [28,29].

The Asian water monitor (*Varanus salvator*) is a large varanid lizard native to southern and southeastern Asia. It is one of the most common monitor lizards in Asia. These varanids feed as both scavengers and predators [30,31]. Because these animals can survive in poor environmental conditions, research into the immune and digestive systems of these animals has

attracted increasing interest [32,33]. Moreover, the growing population of Varanus has led to much debate about managing the exploitation of this species. These reptilian-derived compounds have long been widely used for traditional treatments in Asia. Several components derived from reptiles have been experimentally proven, and anticancer research has been performed on different sources, including extracts, crude peptides, serum, bile, and venom [21]. Bioactive peptides are increasingly being considered good drug candidates for cancer therapeutic application because protein and functional peptides have excellent cell permeability, minimal potential to elicit an immune rejection response, low toxicity and are easy to synthesize and modify [21,26,34,35]. Thus, a growing number of antimicrobial peptides (AMPs) and anticancer peptides (ACPs) derived from the bioactive components of monitor-lizards are being investigated [29,36–38]. However, there is still little related research, and research on this topic is at an early stage. Therefore, the objective of this study was to examine the inhibitory effect of protein hydrolysate and peptide extracted from *Varanus salvator* serum on cancer cells *in vitro*. This investigation identified novel anticancer bioactive molecules for the future treatment of human cancer.

## Materials and methods

### Animals and sampling

The Asian water monitor (*Varanus salvator*) (n=21), which weighed more than 5 kg and had a length of more than 100 cm on the snout to anus, was randomly selected. The sex distribution was not restricted. All Varanus individuals had no clinical signs and had good physical condition. The sampling procedure was carried out at the Khao-zon Wildlife Breeding Station in Ratchaburi Province, Thailand. Animal use was approved by the Faculty of Veterinary Science, Mahidol University-Institute Animal Care and Use Committee (FVS-MU-IACUC). (COA. No. MUVS-2020-11-52). The experiments were performed in accordance with approved guidelines.

Peripheral blood samples (10 ml.) was collected from the caudal tail vein of each *Varanus salvator* using an 18-gauge needle into a plain blood collection tube. Whole blood can coagulate by leaving the blood undisturbed at room temperature for 30 minutes. All the samples were handled in the laboratory at 2–8°C. All blood samples (n=21) were centrifuged at 1,526 x g for 10 minutes in a refrigerated centrifuge (4°C). The clot blood was removed, and the serum was collected and stored at -20°C until further analysis.

### Preparation of native peptides and protein hydrolysate

The <3 kDa peptide fraction was isolated from *Varanus salvator* serum by diluting the serum five-fold with 10 mM sodium acetate buffer (pH 4.0). Fifteen milliliter aliquots of this diluted serum were then centrifuged at 3000 x *g* for 30 minutes at 4°C through a 3 kDa molecular weight cutoff (MWCO) membrane (Vivaspin 20, GE Healthcare, Chicago, USA). The flow-through fraction, containing peptides with a molecular weight of <3 kDa, was collected.

The retentate (>3 kDa) was collected, and its protein concentration was determined using the Lowry method with bovine serum albumin (BSA) as a standard [39]. This >3 kDa fraction, dissolved in 10 mM sodium acetate (pH 4.0), was then subjected to enzymatic hydrolysis using porcine pepsin (Sigma-Aldrich, St. Louis, MO, USA) with an activity exceeding 250 units/mg. Hydrolysis was performed at a 1:20 (w/w) enzyme/substrate ratio at 37°C for 16 hours. The reaction was terminated by heating at 100°C for 10 minutes, and the resulting hydrolysate was filtered through a 3 kDa MWCO membrane to obtain the <3 kDa hydrolysate fraction.

## Peptide purification by reverse-phase chromatography

Ten milliliter aliquots of the <3 kDa native peptide fraction, previously centrifuged at 5000 x *g* for 15 minutes at 4°C, were purified by reverse-phase chromatography using a Delta-Pak C18 column (100 Å, 3.9 mm × 150 mm; Interlink Scientific Services Ltd., Kent, UK). The column was equilibrated with 0.1% trifluoroacetic acid (TFA) in water at a flow rate of 1 mL/min. Following sample loading and a wash with 0.1% TFA in water, the unbound (hydrophilic) fraction was collected. The bound (hydrophobic) fraction was then eluted with 0.1% TFA in 100% acetonitrile. All resulting fractions were concentrated using a SpeedVac vacuum concentrator, resuspended in phosphate-buffered saline, and stored at -20°C until evaluation of anticancer activity.

## Anticancer activities of the native peptides and protein hydrolysate

Protein concentrations of all fractions, reconstituted in phosphate-buffered saline, were determined using the Lowry method with bovine serum albumin (BSA) as a standard. The cytotoxicity of all fractions at 100 µg/mL was evaluated against eighteen culture cell lines (A375, CaCO2, CAL27, NCI-H460, HaCaT, HeLa, HCT8, HT29, HepG2, KATO III, MCF-7, MDA-MB-231, MRC5, Raw264.7, SKOV-3, SW620, T47D, and U937), with Vero cells serving as a normal control.

The tested cell lines derived from *Homo sapiens* include: A375 (malignant melanoma), CAL27 (tongue squamous cell carcinoma), NCI-H460 (lung carcinoma, large cell lung cancer), HeLa (cervical cancer caused by HPV-18), HepG2 (hepatocellular carcinoma), KATOIII (gastric carcinoma), CaCO2, HCT-8, HT-29, SW620 (colorectal adenocarcinoma), MCF-7, MDA-MB-231, T47D (mammary gland/breast adenocarcinoma), SKOV-3 (ovarian adenocarcinoma), U-937 (histiocytic lymphoma), HaCaT (immortalized keratinocyte) and MRC-5 (normal lung fibroblast cell line). The tested cell lines derived from murine was Raw264.7 (Abelson murine leukemia virus-induced tumor monocyte/macrophage). The Vero cells (control) was normal kidney cell line from *Cercopithecus aethiops*.

All cell lines were obtained from the American Type Culture Collection (ATCC; Manassas, VA, USA). They were cultured in minimum essential medium (MEM) (Sigma) supplemented with 10% heat-inactivated fetal bovine serum (FBS) (Gibco), 0.01 bovine insulin, 100 U/ml penicillin and 100 µg/ml streptomycin. The medium was sterilized by passing through a 0.22 µm filter and stored at 4°C until use. The cells were maintained in a humidified incubator with 5% $CO_2$ and 95% air at 37°C as described previously [40]. Cell viability was determined by the MTT assay (Sigma 1350380). Briefly, 10 µl of MTT solution (5 mg/ml in phosphate-buffered saline) was added to each well, and the plates were incubated at 37°C for 2 h. The formazan crystals were dissolved in 100 µl of dimethyl sulfoxide (DMSO), and the absorbance was measured at 590 nm.

## LC−MS analysis

The hydrophobic fractions of the <3 kDa native peptide were analyzed using an Ultimate 3000 Nano/Capillary LC System (Thermo Scientific, UK) coupled to a Hybrid Quadrupole Q-Tof Impact II™ mass spectrometer (Bruker Daltonics) equipped with a nanocapture spray ion source. The 100 ng of each sample was loaded onto a µ-Precolumn C18 Pepmap (300 µm i.d. x 5 mm, 5 µm, 100 Å) and separated on a nanoViper Acclaim PepMap RSLC C18 column (75 µm I.D. × 15 cm, 2 µm, 100 Å) at 60°C. A 30-min linear gradient of 5–55% mobile phase B (0.1% formic acid in 80% acetonitrile) in mobile phase A (0.1% formic acid in water) was applied at 0.30 µL/min. Mass spectra (MS) and tandem mass spectra (MS/MS) were acquired in positive-ion mode (m/z 150–2200) at 2 Hz with 1.6 kV electrospray ionization voltage and

~50 L/h nitrogen drying gas flow. Collision energy was 10 eV (m/z-dependent), and nitrogen was the collision gas for CID. Analyses were performed in triplicate. The amino acid sequence of each peptide in individual samples was determined using MaxQuant 2.1.0.0 and the Uni-Prot *Varanus salvator* database [41]. Then the peptide sequences were studied *in silico*.

### Anticancer peptide (ACP) prediction

The FASTA files of 432 *V. salvator* peptides were submitted to the web server Anti-Cancer Peptide and Anti-Hypertensive Peptide Prediction (ACHP at http://118.178.58.31:9801/). The ACHP algorithm includes three algorithms, namely, support vector machine (SVM), K-nearest neighbor (KNN) and random forest (RF), to train the model, and redundant features are removed using a support vector machine based on the recursive feature elimination method (SVM-RFE). The predicted functional peptides were classified as anticancer peptide (ACP) or antihypertensive peptide (AHP) [42]. The various physicochemical properties, such as molecular weight, isoelectric point (pI), hydrophobicity, and hydrophilicity, were also calculated using the https://web.ExPASy.org/cgi-bin/compute_pi/pi_tool and Peptide2.com for the Peptide Hydrophobicity/Hydrophilicity Analysis Tool (https://www.peptide2.com/). Some selected peptides were modeled using the PEP-FOLD Peptide Structure Prediction Server (https://bioserv.rpbs.univ-paris-diderot.fr/services/PEP-FOLD3/) followed by PyMOL (https://pymol.org/) for 3D structure visualization.

### Statistical analysis

All *Varanus salvator* serum samples in this study were obtained from randomized animals of different sexes, ages and sizes (n=21). The data were checked for a normal distribution and homogeneity of variance. Therefore, nonparametric statistical analysis should be appropriate for this experimental design, in which the central tendency of the dependent variables is presented as medians. The median inhibition values of freely distributed continuous variables between peptides and protein hydrolysate, as well as between the C18-bound fraction (hydrophobic) and the C18-unbound fraction (hydrophilic), were compared using the Wilcoxon rank-sum test. For the continuous variables, a one-sample t test was used to compare the inhibition mean values with a hypothetical cutoff value (k = 70). PASW Statistics for Windows, version 18.0 [43] was used for all analyses. A p value < 0.05 was considered to indicate statistical significance.

### Results

**N**ative peptides and protein hydrolysates derived from Varanus serum produced some results on tested cell lines. The 16 out of 18 cell lines, including A375, CaCO2, CAL27, HaCaT, HeLa, HT29, HepG2, KATO III, MCF-7, MDA-MB-231, MRC5, Raw264.7, SKOV-3, SW620, T47D and U937 cells, were inhibited but Vero cells (control) were less inhibited. However, NCI-H460 and HCT8 were not affected (Table 1). The inhibitory effects of native peptides derived from the serum of 21 *Varanus salvator* strains on 18 types of culture cell lines and Vero cells (control) are shown in S1 Table.

The protein hydrolysates produce a slightly inhibitory effect on CaCO2, CAL27 and SKOV-3 cells. Nine tested cell lines had got 0% inhibition and Vero cells (control) were less inhibited (Table 1). The inhibitory effects of protein hydrolysates derived from the serum of 21 *Varanus salvator* strains on 18 types of culture cell lines and Vero cells (control) are shown in S2 Table.

Except for NCI-H460 and HCT8 cells, which were unaffected by native peptides and protein hydrolysates, native peptides had a significantly greater inhibitory effect on 17 culture cell lines than protein hydrolysates (p value = 0.00) (Table 1).

**Table 1. Inhibitory effect of native peptides and protein hydrolysates derived from the serum of *Varanus salvator* on 18 types of culture cell lines and Vero cells (control) (n=21).**

| cell lines | peptides | protein hydrolysates | p value |
|---|---|---|---|
| | % inhibition | | |
| A375 | 56.80 [a] | 5.40 [b] | 0.000 |
| CaCO2 | 101.40 [a] | 19.30 [b] | 0.000 |
| CAL27 | 100.40 [a] | 12.00 [b] | 0.000 |
| NCI-H460 | 0.00 | 0.00 | 0.317 |
| HaCaT | 95.30 [a] | 4.80 [b] | 0.000 |
| HeLa | 26.40 [a] | 0.00 [b] | 0.000 |
| HCT8 | 0.00 | 0.50 | 0.208 |
| HT29 | 81.90 [a] | 3.20 [b] | 0.000 |
| HepG2 | 83.60 [a] | 0.00 [b] | 0.000 |
| KATO III | 57.30 [a] | 0.00 [b] | 0.000 |
| MCF-7 | 101.40 [a] | 0.00 [b] | 0.000 |
| MDA-MB-231 | 30.60 [a] | 0.00 [b] | 0.000 |
| MRC5 | 90.50 [a] | 0.00 [b] | 0.000 |
| Raw264.7 | 64.90 [a] | 0.00 [b] | 0.000 |
| SKOV-3 | 52.90 [a] | 23.60 [b] | 0.000 |
| SW620 | 50.60 [a] | 4.40 [b] | 0.000 |
| T47D | 44.40 [a] | 0.00 [b] | 0.000 |
| U937 | 99.30 [a] | 0.10 [b] | 0.000 |
| Vero | 7.80 [a] | 3.10 [b] | 0.000 |

[a,b]Median values with different superscript letters in the same row are significantly different (Wilcoxon test, p value < 0.05)

When the inhibition-mean values were compared with a hypothetical cutoff value (k = 70), the Varanus serum peptides were found to be highly effective against 8 of the 18 culture cell lines (CaCO2, CAL27, HaCaT, HT29, HepG2, MCF-7, MRC5, U937) at this threshold (Fig 1).

Native peptides exhibited greater anticancer activity than protein hydrolysates and were subsequently fractionated by stepwise reverse-phase column chromatography. The anticancer activities of the C18 bound fraction (hydrophobic) and C18 unbound fraction (hydrophilic) are presented in Table 2, S3 Table, and S4 Table. Most of the hydrophilic fractions (C18 unbound) had greater inhibitory effects than the hydrophobic fractions (C18 bound) on the culture cell lines.

The prediction of ACHP activity indicated that 159 out of 432 Varanus peptides exhibited potential anticancer efficacy, with probabilities ranging from 86.92% to 50.10%. Notably, one-fourth of these ACPs had a probability exceeding 70%. The top 20 ranked ACPs are listed in Table 3, while their structural representations are shown in Fig 2. Most peptides consisted of 20 or more residues, except for MQLSDNFTKIMLTT, which contained 14 residues. However, peptide length did not influence their secondary structure. The majority of these peptides had low isoelectric points (pIs), with the exception of PLFQKWLLHMLQDYRFRPYSARIW (10.27), GSTDKSPWCATTSNYDRKWKPCA (8.82), ATWQKMAPMALLLATWNLIPT (8.8), FGFKFYNRENFWSQIGSSWT (8.59), and AAIMNWKLCAQLAAFCWGSSFM (8.1).

## Discussion

*Varanus salvator*, an Asian water monitor adapted to harsh environments and known to contain bioactive serum components. This study investigated the in vitro anticancer effects of <3 kDa fractions derived from native peptides and protein hydrolysate of *V. salvator* serum.

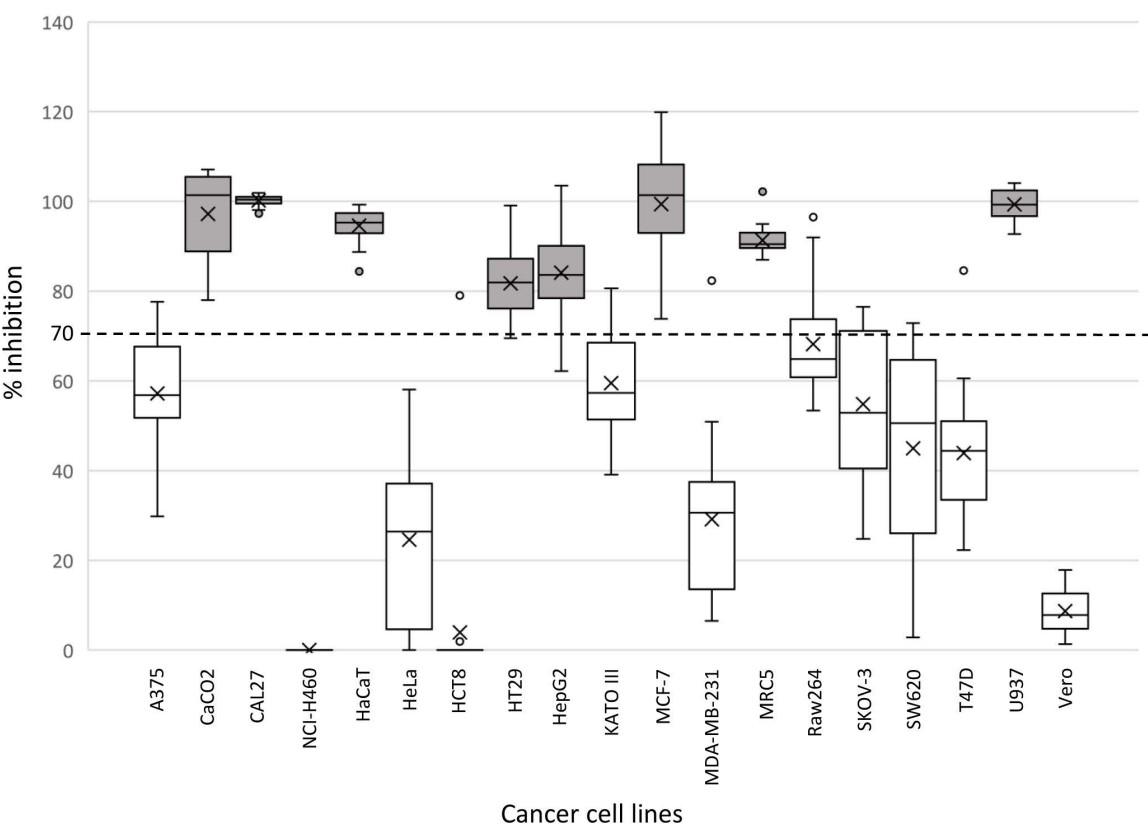

**Fig 1. Native peptides derived from the serum of *Varanus salvator* (n =21) had inhibitory effects on 18 culture cell lines and Vero cells (control) compared to the inhibition cutoff level (70%).** The mean inhibition ± SE are presented in addition to the data distribution. The mean inhibition values, which are represented by graphical gray boxes, were significantly greater than or within the inhibition threshold (one sample t test, p value < 0.05).

Small peptides obtained from *V. salvator* serum demonstrated greater anticancer potential than protein hydrolysates. Hence, the peptides were purified and studied *in vitro*. Compared to those of the C18 Bound fraction, the Unbound fractions exhibited superior efficacy against cancer cell growth. These findings imply that the physicochemical properties of these peptides play roles in anticancer peptide mechanisms [44–49]. However, Unbound fractions show toxicity to Vero cells in this experiment.

To search as much as possibility, the peptide sequences from LC-MS analysis were screening *in silico*. Based on the calculations, several ACPs exhibited high hydrophobicity. These findings might be associated with ACP-cell membrane interactions as well as cell penetration. The predicted ACPs exhibited higher hydrophobicity, whereas the native peptides were hydrophilic. However, some ACPs are hydrophilic or positively charged and might interact well with glycans and the hydrophilic side of phospholipid bilayers on the cell surface, allowing them to more easily penetrate into the cell. Therefore, we plan to synthesize and modify the selected peptides and conduct *in vitro* in further studies. Moreover, our ACPs had α-helical or random coil structures, similar to the findings of previous studies [49] (Fig 2). These highly flexible secondary structures might assist in cell entry of ACPs and consequently enhance their anticancer properties. This finding is consistent with previous studies on positively charged antimicrobial proteins called cationic antimicrobial peptides (CAMPs) derived from Komodo dragon (*Varanus komodoensis*) serum [36].

**Table 2. Inhibitory effect of the C18 bound fraction (hydrophobic) and C18 unbound fraction (hydrophilic) derived from the serum of *Varanus salvator* on 18 types of culture cell lines and Vero cells (n=16).**

| cell lines | hydrophobic | hydrophilic | p value |
|---|---|---|---|
| | **% inhibition** | | |
| A375 | 20.90[a] | 63.80[b] | 0.001 |
| CaCO2 | 98.25 | 99.00 | 0.897 |
| CAL27 | 14.10[a] | 87.80[b] | 0.000 |
| NCI-H460 | 13.05 | 4.45 | 0.433 |
| HaCaT | 18.85[a] | 92.15[b] | 0.000 |
| HeLa | 18.95[a] | 89.10[b] | 0.000 |
| HCT8 | 13.05 | 3.85 | 0.865 |
| HT29 | 14.80 | 8.95 | 0.501 |
| HepG2 | 29.15[a] | 61.90[b] | 0.001 |
| KATO III | 20.00 | 9.95 | 0.796 |
| MCF-7 | 15.20[a] | 52.80[b] | 0.013 |
| MDA-MB-231 | 15.40[a] | 82.60[b] | 0.000 |
| MRC5 | 12.10[a] | 93.70[b] | 0.000 |
| Raw264.7 | 18.70[a] | 88.90[b] | 0.000 |
| SKOV-3 | 43.55 | 39.25 | 0.352 |
| SW620 | 15.30 | 18.65 | 0.301 |
| T47D | 10.35 | 20.7 | 0.084 |
| U937 | 0.00[a] | 98.00[b] | 0.000 |
| Vero | 4.15[a] | 88.10[b] | 0.000 |

[a,b]Median values with different superscript letters in the same row are significantly different (Wilcoxon test, p value < 0.05)

**Table 3. Anticancer peptide prediction of *Varanus salvator* serum using the ACHP web server (http://118.178.58.31:9801/).**

| Sequence | Length | ACP (%) | pI | MW (kDa) | Hydrophobic (%) | Acidic (%) | Basic (%) | Neutral (%) |
|---|---|---|---|---|---|---|---|---|
| FFPNIPNTFEVVFEQHFSSNINYC | 24 | 86.92 | 4.51 | 2.90 | 45.83 | 8.33 | 4.17 | 41.67 |
| AYPWWHMTDYQLCAGILGGGRDTC | 24 | 86.05 | 5.21 | 2.71 | 37.5 | 8.33 | 8.33 | 45.83 |
| AERTMDRWWQYMTLLMTMLG | 20 | 85.73 | 6.11 | 2.53 | 50 | 10 | 10 | 30 |
| ANPTRIIGGQECFEDWHPWL | 20 | 84.63 | 4.65 | 2.37 | 45 | 15 | 15 | 30 |
| AFYQEKDMLSSCRQNSMGHNT | 21 | 84.63 | 6.78 | 2.45 | 23.81 | 9.52 | 14.29 | 52.38 |
| AAIMNWKLCAQLAAFCWGSSFM | 22 | 82.90 | 8.10 | 2.45 | 63.64 | 0 | 4.55 | 31.82 |
| PLFQKWLLHMLQDYRFRPYSARIW | 24 | 82.88 | 10.27 | 3.16 | 54.17 | 4.17 | 20.83 | 20.83 |
| EDSEELYFCPIQLVSFYTTG | 20 | 82.33 | 3.50 | 2.34 | 35 | 20 | 0 | 45 |
| FGFKFYNRENFWSQIGSSWT | 20 | 81.55 | 8.59 | 2.50 | 35 | 5 | 10 | 50 |
| APLGTECDIIGWGETEWVVGSPSD | 24 | 80.23 | 3.39 | 2.52 | 41.67 | 20.83 | 0 | 37.5 |
| DEMRCQYEADLEKNRREVEDWYASQ | 25 | 80.12 | 4.33 | 3.16 | 24 | 32 | 16 | 28 |
| MQLSDNFTKIMLTT | 14 | 79.23 | 5.59 | 1.64 | 42.86 | 7.14 | 7.14 | 42.86 |
| DVFHWYWTAAEFGFHHNDTFHHK | 23 | 78.23 | 6.15 | 2.93 | 39.13 | 13.04 | 26.09 | 21.74 |
| ACAFQMLAYLALLIGYHATVDNS | 23 | 78.22 | 5.08 | 2.48 | 56.52 | 4.35 | 4.35 | 34.74 |
| CWNVVQRKFSNSPCEVYFPRNDTME | 25 | 77.88 | 6.17 | 3.05 | 36 | 12 | 12 | 40 |
| ATWQKMAPMALLLATWNLIPT | 21 | 76.38 | 8.8 | 2.37 | 71.43 | 0 | 4.76 | 23.81 |
| GSTDKSPWCATTSNYDRDRKWKPCA | 25 | 76.12 | 8.82 | 2.87 | 24 | 12 | 20 | 44 |
| AEDNELNNPEGTNWPKYSTSTRDT | 24 | 75.82 | 4.18 | 2.74 | 20.83 | 20.83 | 8.33 | 50 |
| AERYYCKSEEVRALDCARSFSPNI | 24 | 75.63 | 6.28 | 2.81 | 33.33 | 16.67 | 16.67 | 33.33 |
| AEASFMCGATLLNQDWVLTAAHCY | 24 | 75.13 | 4.35 | 2.61 | 50 | 8.33 | 4.17 | 37.5 |

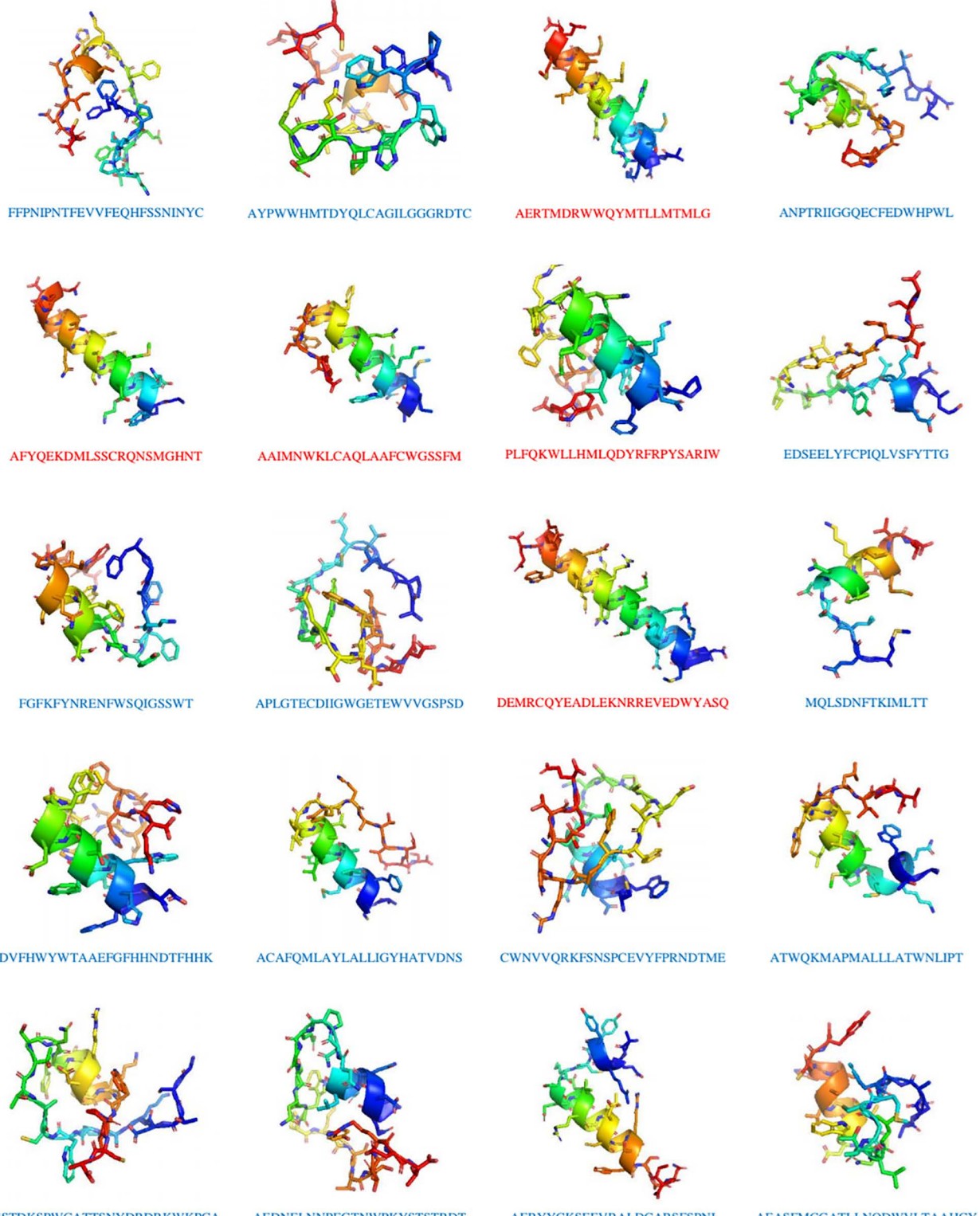

**Fig 2. 3D structures of the top twenty ranked anticancer peptides (ACPs) from *Varanus salvator* serum predicted using the ACHP web server and generated models using PEP-FOLD followed by PyMOL.** The predicted peptides were divided into two conformations: a random coil (blue alphabet amino acid sequence) or an α-helix (red alphabet amino acid sequence).

From this experiment, the native peptides derived from Varanus serum show highly evidence against colorectal adenocarcinoma, squamous cell carcinoma, hepatocellular carcinoma, mammary gland/breast adenocarcinoma and histiocytic lymphoma. However, the mechanism of action is still unknown. The anticancer peptide mechanisms might be classified based on the mechanism of cell entry or their actions. Pore-forming peptides, cell-penetrating peptides, or tumor-targeting peptides could be useful [44]. The ACPs are divided into three types according to their actions: 1) molecular targeted peptides, 2) 'guiding missile' peptides or binding peptides, and 3) cell-stimulating peptides [49]. Moreover, two mechanisms of action of ACP models have been proposed: i) the 'barrel-stave' model proposed by Ehrenstein and Lecar in 1977 and ii) the 'carpet' model proposed by Pouny *et al.* in 1992 [48]. Taken together, the efficacy of these anticancer peptides seems to rely on their intrinsic and extrinsic factors, such as their secondary structure, mode of action against cancer cells, physicochemical properties, and cell entry capacity. Future prospects, additional studies of the ACP candidates and their modifications will improve their anticancer efficiency and should focus on the underlying mechanism and clinical application of ACPs.

## Conclusions

In conclusion, native peptides derived from Asian water monitor (*Varanus salvator*) serum seem to have high anticancer potential for many cancer cells, such as those of colorectal adenocarcinoma, squamous cell carcinoma, hepatocellular carcinoma, mammary gland/breast adenocarcinoma and histiocytic lymphoma. Twenty potential anticancer peptides (ACPs) were identified with a high ACP ranking (more than 75%). The next step is to synthesize these peptides and subsequently study them *in vitro* to identify the most effective ACPs. Further modifications may be required to enhance the anticancer efficacy and reduce toxicity to Vero cells. This study serves as a starting point for the future development of novel anticancer drugs derived from Varanus serum peptides.

## Supporting information

**S1 Table. Inhibitory effect on 18 types of culture cell lines and Vero cells (control) of native peptides derived from serum of *Varanus salvator* (VS) (n=21).**
(PDF)

**S2 Table. Inhibitory effect on 18 types of culture cell lines and Vero cells (control) of protein hydrolysates derived from serum of *Varanus salvator* (VS) (n=21).**
(PDF)

**S3 Table. Inhibitory effect on 18 types of culture cell lines and Vero cells of C18 Bound fraction (hydrophobic) derived from serum of *Varanus salvator* (VS) (n=16).**
(PDF)

**S4 Table. Inhibitory effect on 18 types of culture cell lines and Vero cells of C18 Unbound fraction (hydrophilic) derived from serum of *Varanus salvator* (VS) (n=16).**
(PDF)

## Acknowledgments

The author would like to thank Mr. Pakpoom Aramsirirujiwet for sampling assistance at Khao-zon Wildlife Breeding Station and the Department of National Park Wildlife and Plant Conservation for providing various facilities.

## Author contributions

**Conceptualization:** Jitkamol Thanasak, Sittiruk Roytrakul, Rudee Surarit.

**Data curation:** Jitkamol Thanasak, Sittiruk Roytrakul, Waraphan Toniti, Janthima Jaresitthikunchai, Narumon Phaonakrop, Siriwan Thaisakun, Sawanya Charoenlappanit.

**Formal analysis:** Jitkamol Thanasak, Sittiruk Roytrakul, Waraphan Toniti, Surasak Jittakhot.

**Funding acquisition:** Jitkamol Thanasak.

**Investigation:** Jitkamol Thanasak, Sittiruk Roytrakul, Rudee Surarit, Waraphan Toniti, Wanna Sirimanapong.

**Methodology:** Jitkamol Thanasak, Sittiruk Roytrakul, Rudee Surarit, Waraphan Toniti, Wanna Sirimanapong, Janthima Jaresitthikunchai, Narumon Phaonakrop, Siriwan Thaisakun, Sawanya Charoenlappanit.

**Project administration:** Jitkamol Thanasak.

**Resources:** Jitkamol Thanasak, Sittiruk Roytrakul.

**Software:** Waraphan Toniti.

**Supervision:** Jitkamol Thanasak, Sittiruk Roytrakul, Rudee Surarit.

**Validation:** Jitkamol Thanasak, Sittiruk Roytrakul, Rudee Surarit, Waraphan Toniti, Surasak Jittakhot.

**Visualization:** Waraphan Toniti.

**Writing – original draft:** Jitkamol Thanasak, Sittiruk Roytrakul, Waraphan Toniti.

**Writing – review & editing:** Jitkamol Thanasak, Sittiruk Roytrakul, Waraphan Toniti.

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
