## [Decision Letter · Decision Letter 0]

7 Jan 2025

PONE-D-24-53048Anticancer Properties of Peptides and Protein hydrolysates Derived from Asian Water Monitor (Varanus salvator) serumPLOS ONE

Dear Dr. Thanasak,

Thank you for submitting your manuscript to PLOS ONE. After careful consideration, we feel that it has merit but does not fully meet PLOS ONE’s publication criteria as it currently stands. Therefore, we invite you to submit a revised version of the manuscript that addresses the points raised during the review process.

We look forward to receiving your revised manuscript.

Kind regards,

Ruo Wang

Academic Editor

PLOS ONE

Journal Requirements:

“The research project was supported by Mahidol University.”

**Additional Editor Comments:**

Please note that requests for additional citations from some reviewers do not constitute a factor in the editorial decision on whether a manuscript should be accepted. Therefore, authors do not need to refer to these comments unless absolutely necessary.

Reviewers' comments:

Reviewer's Responses to Questions

**Comments to the Author**

1. Is the manuscript technically sound, and do the data support the conclusions?

Reviewer #1: No

Reviewer #2: Yes

Reviewer #3: Partly

2. Has the statistical analysis been performed appropriately and rigorously? 

Reviewer #1: N/A

Reviewer #2: Yes

Reviewer #3: I Don't Know

3. Have the authors made all data underlying the findings in their manuscript fully available?

Reviewer #1: No

Reviewer #2: Yes

Reviewer #3: Yes

4. Is the manuscript presented in an intelligible fashion and written in standard English?

Reviewer #1: Yes

Reviewer #2: Yes

Reviewer #3: Yes

5. Review Comments to the Author

Reviewer #1: 1. The manuscript does not provide sufficient evidence to explain the molecular mechanisms by which the peptides exert their anticancer effects. Mechanistic studies, such as those exploring apoptosis induction, cell cycle arrest, or the modulation of key signaling pathways (e.g., PI3K/AKT, MAPK), are necessary to move beyond descriptive findings and contribute to a deeper scientific understanding.

2. The study identifies 159 peptides with potential anticancer activity based on computational predictions but lacks thorough experimental validation. High-confidence predictions must be supported by biochemical assays, such as binding affinity measurements or functional tests in cancer models, to ensure credibility and reliability.

Reviewer #2: This manuscript explores the anticancer potential of crude peptides and protein hydrolysates from Varanus salvator serum across various cancer cell lines. It demonstrates promising inhibitory activity, especially from hydrophilic peptides, with computational insights suggesting novel anticancer peptides. While intriguing, further studies on specificity, safety for normal cells, and peptide optimization are crucial for therapeutic application. On reviewing the manuscript, I found the study to be interesting and potentially suitable for acceptance. However, the manuscript cannot be accepted in its current form. Major revisions are required before it can be considered for acceptance.

1. The manuscript requires improvement in terms of language and grammar. There are several instances of unclear phrasing, and grammatical errors that hinder readability

2. The introduction should include a paragraph discussing the current state of cancer and its related mortality rates.

3. The authors must discuss the future prospects and limitations of their study.

4. In the introduction section, the authors are advised to include the following references to support their statements and provide a stronger foundation for their study.

https://doi.org/10.1016/j.ejmech.2024.117088

https://doi.org/10.1002/slct.202101853

https://doi.org/10.1039/C3RA45749D

https://doi.org/10.1016/j.phymed.2024.155972

https://doi.org/10.3389/fphar.2023.1168566

https://doi.org/10.3389/fphar.2023.1231450

Reviewer #3: Manuscript Number: PONE-D-24-53048

Anticancer Properties of Peptides and Protein hydrolysates Derived from Asian Water

Monitor (Varanus salvator) serum

The concept of “SAFE” needs explanation. Are Vero cells the test control for “SAFE/ SAFETY”?

37 ….they are safe for normal cells….

82 ….peptides are safe drugs….

304 ….maintain its safety in normal….

The concept of “normal cell” must be clarified. Are “normal” and/or “control” a non-cancerogenic cell? Vero cells are a continuous cell line, a kind of immortalized cells, but are not Vero “cancer cells”?

HeLa is an immortalized cancer cell line.

The cell line MRC is in the text as cancer cell but in the Lines 127, 128, 209, and 241 is ….MRC-5 = Homo sapiens, NORMAL, Lung.…

205/237 ….HaCaT = Homo sapiens, keratinocyte….

Is keratinocyte cancer or normal cell?

268 ….the peptides were purified and studied IN SILICO

Definition of IN SILICO.

Serum samples from Varanus salvator were ultra filtered with 3 kDa membrane with and without purification by reverse phase chromatography in C18, peptides tested with cancer and normal cells. I think this is “in vitro”.

Table 1, lines 203 to 213 and Table 2, l7774ines 235 to 245 are many repetitions, that explanation could be in the Material and Methods, as a table.

Homo sapiens has 32 repetitions!

Vero cells (control) have 9 repetitions.

colon, colorectal adenocarcinoma; 8 repetitions

adenocarcinoma 22 repetitions

Table 1 and Figure 1 have some difference in the numbers. The Figure 1 do not need the numbers, they are represented graphically, and the important values are in the Table 1. Figure 1 has many numbers, it is polluted.

Lines 214 to 217, is Raw264.7 inhibited by the peptides? The limit is 70% inhibition, Raw264.7 has 64.9. In Figure 1, Raw264.7 is under dashed line 70%.

91 Materials and Methods

92 Animals and sampling

103 .... centrifuged at 2,500 RPM for 10 ...

RPM units convert to centrifugation speed (× g)

106 Preparation of crude peptides and protein hydrolysate

107 .... fivefold with 0.5 M sodium acetate and….

Sodium acetate pH??

108 ….(Vivaspin 20, GE Healthcare, Chicago, USA)….

(Vivaspin 20 spin columns, Cytiva)

It is necessary indication of centrifugation speed (× g), time, temperature °C,

range of applied diluted sample volume.

112 and 113 Protein hydrolysis: buffer conditions, pH, composition,

concentration, and specific activity of pepsin for hydrolysis.

Sigma–Aldrich has pepsin from ≥400 up to ≥3,200 U/mg protein.

116 Peptide Purification by Reverse-Phase Chromatography

The peptides samples were applied without any adjustment, preparation? like

centrifugation or micro filtration? addition of concentrate trifluoroacetic

acid?

Volume or protein amount of native and hydrolysate peptide fraction applied

to the hydrophobic Delta-Pak C18.

122 ….with 0.1% TFA in acetonitrile….

% acetonitrile? linear or step gradient?

123 & 124 ….Unbound and bound fractions… were evaluated for their anticancer activity.

The fraction from C18 were used directly with Vero and cancer cells without

any treatment? pH adjustment? Elimination of acetonitrile? Protein

concentration or dilution? Buffer change? Storage condition of the fraction?

125 Anticancer activities of the crude peptides and protein hydrolysate

There are not explanations for the condition of the purified fraction in the

cytotoxicity test: protein concentration, buffer etc.

139 LC‒MS analysis

140 The hydrophobic fractions ….

were hydrophilic fractions (bound to C18) analysed by LC-MS analysis?

145 ~154 It is too long, reduce.

183 Results

184/185 ….protein hydrolysates….produced some results in cancer cell lines.

Table I showed that protein hydrolysates HAVE NOT INHIBITORY EFFECT over cancer cells. Nine cancer cell lines have 0% inhibition, and the best inhibition effect has only 23,6%.

225 By reverse-phase chromatography, the PEPTIDES were purified.

230 Table 2 there are not indication the origin of the pure hydrophobic and hydrophilic after the RP-HPLC Delta-Pak C18.

Lines 117/118 Both the <3 kDa native peptide fraction and the <3 kDa hydrolysate fraction were purified by reverse-phase high-performance liquid chromatography (RP-HPLC)

Table 2 needs separation of the results from native peptide and hydrolysate fraction post-C18. Protein hydrolysate fraction has not anticancer activity but its hydrophilic (C18 unbound) may be has.

Hydrophilic peptides have inhibitory activity against cancer cells, but also hydrophilic peptides have an effect, 88.1% inhibition, against Vero normal cells!

The test for safety with Vero cells showed that hydrophilic peptides are inhibitors for this line cell.

Anticancer drugs must be toxic for tumoral cells but atoxic for normal cells.

It is confusing the purified peptide fraction origin of Table 2, crude peptides or hydrolyzed protein.

247 The prediction of ACHP activity showed that 159 of the 432….

It is written in line 140 The HYDROPHOBIC FRACTIONS of both the <3 kDa native peptide and the <3 kDa hydrolysate….LC-MS analysis

The hydrophobic fractions of C18 are not active against cancer cells, Lines 31/32; 227/228 and 268/269/270.

268 ….Hence, the crude/natural peptides were purified and studied in silico “in vitro”.

In the manuscript is not clear the flow of serum proteins, peptides, flow of raw or purified samples to in vitro test like cell growth inhibition, sample fraction purify by C18, purified fraction that are analysed by LC‒MS analysis.

272 Based on the calculations, several ACPs exhibited high hydrophobicity

Your experimental data showed that hydrophilic peptides are better for inhibitory activities with cancer cells

6. PLOS authors have the option to publish the peer review history of their article (what does this mean? ). If published, this will include your full peer review and any attached files.

**Do you want your identity to be public for this peer review?** For information about this choice, including consent withdrawal, please see our Privacy Policy .

Reviewer #1: No

Reviewer #2: No

Reviewer #3: **Yes: ** Joaquin Cabrera-Crespo

---

## [Author Response · Author response to Decision Letter 1]

12 Feb 2025

Dear reviewers

According to some points raised by the academic reviewers to the manuscript entitled “Anticancer Properties of Peptides and Protein hydrolysates Derived from Asian Water Monitor (Varanus salvator) serum”. Please see the responses (the answers in blue) to each point in Response to Reviewers file.

Thank you to the reviewers for highlighting the questions. We sincerely appreciate the reviewer’s feedback, as it has assisted us in improving the clarity and completeness of our data presentation and conclusions.

Your sincerely,

Jitkamol Thanasak, Corresponding author

---

## [Decision Letter · Decision Letter 1]

10 Mar 2025

Anticancer Properties of Peptides and Protein hydrolysates Derived from Asian Water Monitor (Varanus salvator) serum

PONE-D-24-53048R1

Dear Dr. Thanasak,

We’re pleased to inform you that your manuscript has been judged scientifically suitable for publication and will be formally accepted for publication once it meets all outstanding technical requirements.

Kind regards,

Ruo Wang

Academic Editor

PLOS ONE

Additional Editor Comments (optional):

Reviewers' comments:

Reviewer's Responses to Questions

**Comments to the Author**

1. If the authors have adequately addressed your comments raised in a previous round of review and you feel that this manuscript is now acceptable for publication, you may indicate that here to bypass the “Comments to the Author” section, enter your conflict of interest statement in the “Confidential to Editor” section, and submit your "Accept" recommendation.

Reviewer #2: All comments have been addressed

Reviewer #3: All comments have been addressed

2. Is the manuscript technically sound, and do the data support the conclusions?

Reviewer #2: Yes

Reviewer #3: Yes

3. Has the statistical analysis been performed appropriately and rigorously? 

Reviewer #2: Yes

Reviewer #3: I Don't Know

4. Have the authors made all data underlying the findings in their manuscript fully available?

Reviewer #2: Yes

Reviewer #3: Yes

5. Is the manuscript presented in an intelligible fashion and written in standard English?

Reviewer #2: Yes

Reviewer #3: Yes

6. Review Comments to the Author

Reviewer #2: The authors have properly addressed all comments, and the manuscript is now suitable for publication.

Reviewer #3: Acepted for publication but can have minor improvements

Varanus salvator in Title, first in Abstract and first in Introduction, next citations V. salvator

First time “python Malayopython reticulatus” next citations M. reticulatus

Vivaspin 20, GE Healthcare Cytiva

RESULTS

Only eight of the tested cell lines had a 70% inhibition: CaCO2, CAL27, HaCaT, HT29, HepG2, MCF-7, MRC5, U937 with native peptides and no inhibition with protein hydrolysate. Hydrolysate has no activity; the best has 23.6% and 7 are 0%. It is too long an explanation for no activity, it is confusing.

I am not in agreement with this sentence

“Native peptides exhibited GREATER ANTICANCER ACTIVITY THAN PROTEIN HYDROLYSATES and were subsequently fractionated by stepwise reverse-phase column chromatography”

Hydrophilic fraction has activity against 8 cell lines: CaCO2, CAL27, HaCaT, HeLa, MDA-MB-231, MRC5, Raw264.7, U937, but also against normal Vero cell. Hydrophobic has not anti-cancer activity

Notably, one-fourth of these ACPs had a probability exceeding 70% [Results]

70% or 75%?

…top twenty had a probability of more than 75%. [Abstract]

I could not find any results about LC‒MS analysis [Mass Spectra] for the hydrophobic .fractions of both the <3 kDa native peptide and the <3 kDa hydrolysate.

7. PLOS authors have the option to publish the peer review history of their article (what does this mean? ). If published, this will include your full peer review and any attached files.

**Do you want your identity to be public for this peer review?** For information about this choice, including consent withdrawal, please see our Privacy Policy .

Reviewer #2: No

Reviewer #3: **Yes: ** joaquin Cabrera-Crespo

---

## [Editor Report · Acceptance letter]

PONE-D-24-53048R1

PLOS ONE

Dear Dr. Thanasak,

I'm pleased to inform you that your manuscript has been deemed suitable for publication in PLOS ONE. Congratulations! Your manuscript is now being handed over to our production team.

Kind regards,

on behalf of

Dr. Ruo Wang

Academic Editor

PLOS ONE